# Unveiling the Role of HGF/c-Met Signaling in Non-Small Cell Lung Cancer Tumor Microenvironment

**DOI:** 10.3390/ijms25169101

**Published:** 2024-08-22

**Authors:** Shuxi Yao, Xinyue Liu, Yanqi Feng, Yiming Li, Xiangtian Xiao, Yuelin Han, Shu Xia

**Affiliations:** Department of Oncology, Tongji Hospital, Tongji Medical College, Huazhong University of Science and Technology, Wuhan 430030, China; yaoshuxi@hust.edu.cn (S.Y.); liuxinyuedoctor01@hust.edu.cn (X.L.); d202382250@hust.edu.cn (Y.F.); d202282140@hust.edu.cn (Y.L.); m202376505@hust.edu.cn (X.X.); u201910354@hust.edu.cn (Y.H.)

**Keywords:** HGF, c-MET, tumor microenvironment, immunotherapy, NSCLC

## Abstract

Non-small cell lung cancer (NSCLC) is characterized by several molecular alterations that contribute to its development and progression. These alterations include the epidermal growth factor receptor (EGFR), anaplastic lymphoma kinase (ALK), human epidermal growth factor receptor 2 (HER2), and mesenchymal–epithelial transition factor (c-MET). Among these, the hepatocyte growth factor (HGF)/c-MET signaling pathway plays a crucial role in NSCLC. In spite of this, the involvement of the HGF/c-MET signaling axis in remodeling the tumor microenvironment (TME) remains relatively unexplored. This review explores the biological functions of the HGF/c-MET signaling pathway in both normal and cancerous cells, examining its multifaceted roles in the NSCLC tumor microenvironment, including tumor cell proliferation, migration and invasion, angiogenesis, and immune evasion. Furthermore, we summarize the current progress and clinical applications of MET-targeted therapies in NSCLC and discuss future research directions, such as the development of novel MET inhibitors and the potential of combination immunotherapy.

## 1. Introduction

Globally, lung cancer is the leading cause of cancer-related deaths. NSCLC accounts for 80% of all lung cancers and typically has a poor prognosis [1]. Due to the significant genetic and cellular heterogeneity of NSCLC [2], determining a tumor’s histological type and associated molecular changes is crucial for optimizing treatment. Oncogenic alterations in EGFR, ALK, MET, and other kinases drive NSCLC development [3]. Previous studies have confirmed that MET is a significant driver of mutation in NSCLC, activating signaling pathways downstream of PI3K/AKT, Ras-MAPK, and STAT3 (signal transducer and activator of transcription 3), which promote cancer proliferation, migration, and invasion [4]. Furthermore, MET amplification contributes to EGFR-TKI resistance [5]. The implementation of MET inhibition in the clinical management of NSCLC represents an increasingly significant area of research. Several MET inhibitors have been developed that that demonstrate considerable anti-tumor efficacy, particularly in cases with MET exon 14 skipping mutations (METex14) or MET amplification [6]. In the GEOMETRY mono-1 clinical trial, capmatinib (MET TKI) demonstrated clinically meaningful activity, achieving an overall response rate (ORR) of 68% in patients with the METex14, and ORRs of 40% in treatment-naive patients and 29% in those who had received prior therapy for patients with MET amplification and gene copy number (GCN) ≥10 [7]. Preliminary results from the INSIGHT 2 clinical trial indicate that combining tepotinib with osimertinib yields a 55% ORR in patients with EGFR-mutated and MET-amplified NSCLC, suggesting that the tepotinib may have potential efficacy in overcoming EGFR-TKI resistance [8]. However, acquired resistance to MET inhibitors remains a challenge, often due to secondary MET mutations, amplifications, or activation of alternative pathways like EGFR, HER3, and PI3K [9,10,11].

Research in recent years has revealed that the tumor microenvironment (TME) plays a crucial role in the development of drug resistance in tumors [12]. Cancer cells can resist conventional therapies (e.g., chemotherapy and radiotherapy) and emerging targeted therapies and immunotherapies by altering components and signaling pathways within the TME. Immunotherapy has become a highly promising treatment [13]. However, its effectiveness is significantly hindered by various immunosuppressive factors present in the TME. Tumor cells can reshape their microenvironment to form robust immunosuppressive networks, thereby limiting the efficacy of T cells in targeting and destroying tumor cells [14]. Therefore, an in-depth understanding of the TME and tumor-induced immunosuppression mechanisms is essential for improving cancer immunotherapy. This knowledge will facilitate the development of new molecular interventions to be used in combination with various cancer immunotherapies, thereby enhancing treatment specificity and efficacy.

In the TME, c-MET and hepatocyte growth factor (HGF) play a crucial role [15]. This pathway enhances tumor cell proliferation and survival, supporting rapid tumor growth [16]. Additionally, HGF/c-MET signaling promotes the migration and invasive capabilities of tumor cells, facilitating local invasion and distant metastasis [17]. Furthermore, the HGF/c-MET pathway stimulates angiogenesis, ensuring an adequate blood and nutrient supply to the tumor, thus fostering further growth and expansion [18]. Importantly, the HGF/c-MET signaling pathway is also instrumental in tumor immune escape. By altering the composition and function of immune cells in the TME, HGF/c-MET signaling can inhibit the anti-tumor immune response, allowing tumor cells to evade immune system attacks. Therefore, an in-depth study of HGF/c-MET mechanisms in the TME will enhance our biological perspective on tumors and provide a critical theoretical basis and potential targets for developing novel anti-cancer therapies.

## 2. Structure and Function of HGF/c-MET Signaling

Hepatocyte growth factor/scatter factor (HGF/SF) is a paracrine signaling molecule generated and released by mesenchymal stromal cells, serving as the sole ligand for c-MET [19]. HGF is a heterodimer generated by proteolytic cleavage of a single precursor protein, initially synthesized as an inactive single-chain precursor (pro-HGF). Through protease cleavage, pro-HGF is converted into an active heterodimer consisting of a heavy chain (α-chain, approximately 57 kDa) and a light chain (β-chain, approximately 26 kDa), which are linked by disulfide bonds to form an active protein of about 82 kDa [20]. HGF contains several functional structural domains. The N-terminus consists of an N-terminal fragment (N-terminal domain, N) and four tandem kringle domains (K1–K4), which play crucial roles in receptor binding and activation [21]. The kringle domain is a protein secondary structure with a ring-like formation typically involved in protein–protein interactions. The light chain structural domain (β-chain) of HGF contains a serine protease-like domain, which, although not enzymatically active, is essential for the binding and signaling between HGF and c-MET.

c-MET, also referred to as tyrosine protein kinase Met, is a receptor tyrosine kinase (RTK). Its single-chain precursor, pro-MET, is processed by protein hydrolysis into a mature disulfide-linked heterodimer that becomes a functional c-MET receptor. This receptor is found on the surface of epithelial and endothelial cells and comprises several distinct regions: an extracellular portion, a transmembrane domain, an intracellular portion, and a carboxy terminal domain (C-terminal) [22]. The extracellular part includes the immunoglobulin-plexin-transcription (IPT) domain, plexin-semaphorin-integrin (PSI) domain, and the semaphorin (Sema) domain, which facilitate HGF binding [23]. The intracellular region includes the juxtamembrane (JM) domain, the catalytic domain, and a docking site essential for signal transduction [24].

When HGF binds to c-MET, it causes c-MET to homodimerize and its intracellular kinase domains to undergo trans-phosphorylation., which occurs mainly at several tyrosine residues in the intracellular tail of MET, including Y1234 and Y1235 (key sites in the activation loop) and Y1349 and Y1356 (multifunctional docking sites) [25]. Phosphorylated tyrosine residues (especially Y1349 and Y1356) provide binding sites for a range of signaling molecules [26], including GRB2 (growth factor receptor-bound protein 2), GAB1 (GRB2-associated-binding protein 1), PI3K (Phosphatidylinositol 3-kinase), PLCγ (Phosphatidylinositol-specific phospholipase Cγ), and STAT3. These signaling molecules recognize and bind to phosphorylated tyrosine residues through the specific SH2 domain (Src homology 2 structural domain) [27]. Signaling molecules bound to MET receptors further activate various downstream signaling pathways, including Ras-MAPK, PI3K-AKT, STAT, β-strand integrins, and GAB1-mediated signaling pathways. These pathways influence a range of biological processes, encompassing cell proliferation, migration, survival, morphological changes, and differentiation [17,28] (Figure 1). 

Notably, there is significant crosstalk between the c-MET receptor and other receptor tyrosine kinases (RTKs). The c-MET receptor interacts with other RTKs, primarily through heterodimeric complex formation, crucial for signal transduction [29]. In lung cancer, MET can form heterodimers with RTKs such as EGFR, HER2, HER3, and RET. These heterodimerization chaperones of MET have been reported to be highly phosphorylated in MET amplification-positive NSCLC and dependent on MET kinase activity [30]. Similarly, RON, which belongs to the same subfamily as MET, has been demonstrated to form heterodimers with c-MET, thereby initiating transphosphorylation. This interaction is crucial for maintaining the MET-amplified non-small cell lung cancer (NSCLC) oncogenic phenotype [31,32]. Beyond heterodimerization-activated RTKs, additional mechanisms exist through which MET and other RTKs interact. The AXL receptor tyrosine kinase (AXL), a member of the TAM family, is recognized as a pro-migratory and pro-proliferative protein. A direct interaction between c-MET and AXL has been documented, wherein hepatocyte growth factor (HGF) induces co-polymerization of MET and AXL proteins at the plasma membrane, leading to MET kinase-mediated phosphorylation of AXL [33]. Additionally, there exists lateral signaling between EGFR and c-MET. EGFR ligands promote increased expression of c-MET and phosphorylation of its tyrosine residues, with the c-Src pathway playing a crucial role in mediating EGFR-to-c-MET communication [34]. Beyond its crosstalk with other RTKs, the c-MET receptor can also be activated by G-protein-coupled receptors (GPCRs), potentially through NADPH-induced reactive oxygen species release [35].

Although the HGF/c-MET pathway has critical physiological functions in normal cellular processes, its abnormal activation is strongly linked to the development of NSCLC [36,37].

## 3. HGF/c-Met Signaling in NSCLC

### 3.1. Dysregulated HGF/c-MET Signaling in NSCLC

A variety of mechanisms contribute to the addiction of NSCLC to the HGF/c-MET axis, including MET overexpression and point mutations, amplifications, and fusions of the MET gene.

MET overexpression rates vary from 15% to 70% in NSCLC [38,39,40]. In addition to total protein levels, ligand-activated MET, identified by phosphorylated MET (p-MET), is present in approximately two-thirds of lung cancer samples [38,41]. The cytoplasmic form of p-MET, Y1003, and the nuclear version, Y1365, have been found associated with negative prognoses and shown to be predictive biomarkers for MET inhibitors. MET overexpression was identified using immunohistochemical (IHC) methods and different scoring systems, most commonly staining based on a 0 to 3+ scale.

MET exon 14 skipping mutations (METex14) are a common type of point mutation found in 3–4% of NSCLC patients [42,43,44,45]. There is a strong correlation between the mutation rate and histological subtypes of NSCLC, with sarcomatoid carcinoma having the highest mutation rate (4.9–31%), followed by adenosquamous carcinoma (5%), adenocarcinoma (3%), and squamous cell carcinoma (2%) [46,47,48]. Exon 14 encodes the binding site for the ubiquitin ligase CBL, which mediates MET degradation. The mutation causes a decrease in MET degradation and an increase in stability [49]. It has been demonstrated that patients with METex14 are more likely to benefit from MET TKIs [48,49,50,51,52,53,54,55,56,57]. To date, the identification of MEtex14-mutated tumors has been facilitated by various methodologies, including DNA next-generation sequencing (NGS) platforms, Sanger sequencing, and RNA-based assays such as reverse transcription–polymerase chain reaction (RT-PCR) and RNA-based NGS. Among these, NGS demonstrates the highest sensitivity. This sensitivity is attributed to the flexibility of employing different NGS assays tailored to the analysis of either DNA or RNA extracted from tumor tissue. Specifically, DNA sequencing is adept at identifying variants that modify or eliminate splice sites, while RNA sequencing directly observes the consequences of altered splicing. Notably, RNA sequencing offers superior sensitivity compared to DNA-based NGS in detecting these mutations [49,58,59]. However, the application of NGS for METex14 in routine clinical practice is still constrained by several limitations. Notably, up to 40% of tissue biopsies are inadequate for molecular testing, precluding comprehensive genomic analysis for all patients [60].

Two to five percent of NSCLC cases have been found to have MET amplification [42]. Only 0.34 percent of patients have high-level MET gene amplification. It is possible that high MET gene amplification may act as a carcinogenic driver in these patients, since no other oncogenic driver genes were detected in these patients [61]. Crizotinib is highly effective in treating advanced NSCLC with high levels of MET amplification [62]. Further, MET amplification leads to resistance to EGFR TKIs by activating ERBB3/PI3K/AKT signaling [63]. In clinical practice, alterations in MET copy number can be identified using various assays, including fluorescence in situ hybridization (FISH), quantitative reverse transcription polymerase chain reaction (RT-PCR), and NGS. FISH, in particular, detects MET amplification by assessing the ratio of MET copy number to the copy number of chromosome 7 (CEP7). RT-PCR is a straightforward and efficacious technique. Recognition of MET amplification by NGS is contingent upon the specific platform employed. Nevertheless, the ambiguous definition of MET amplification cutoff values and the variability among observers pose significant challenges to the application of a trial’s findings in routine clinical practice [64]. To enhance the development of personalized treatment strategies, clinicians should integrate multi-dimensional data encompassing gene expression, protein levels, and functional status to achieve a comprehensive understanding of MET’s role. For instance, initial screening for MET overexpression can be conducted using IHC, followed by confirmation of gene amplification through FISH and NGS, which subsequently establishes METamp cutoffs. Additionally, digital pathology minimizes inter-observer variability in the interpretation of FISH results. Data derived from MET analysis can be integrated with other clinical and molecular data to predict patient responses to treatment, thereby enhancing the clinical decision-making process for more precise and accurate care.

Just 0.2% to 0.3% of lung cancer patients exhibit MET fusions [65]. Current MET-fusing genes include TPR, TRIM4, ZKSCAN1, PPFIB1, LRRFIP1, EPS15, DCTN1, PTPRZ1, NTRK1, CLIP2, TFG, and HLA-DRB1 [66]. MET pathway activation can also occur through the upregulation of MET or HGF secretion signals, resulting in tumor transformation, in addition to gene amplification, mutation, or fusion [67,68]. The contemporary techniques for identifying MET gene fusions encompass IHC, FISH, RT-PCR, and NGS. IHC presents challenges in differentiating between fusions and other mutations, with a potential for false positives [69]. While FISH is capable of detecting most MET fusions, it is limited in its ability to identify intragenic fusions. In current clinical practice, NGS has emerged as the predominant method for the identification of MET fusions [70].

To implement precise treatment for MET-altered patients in clinical practice, it is essential to adopt a comprehensive post-technical strategy for MET analysis. Clinicians must not only integrate multiple assays, including immunohistochemistry (IHC), fluorescence in situ hybridization (FISH), and next-generation sequencing (NGS), but also establish and adhere to rigorous testing standards and quality control measures to ensure the consistency and reliability of MET testing. Furthermore, MET analysis data should be integrated with other clinical and molecular datasets to facilitate big data analysis and predictive modeling, potentially enhanced by artificial intelligence.

As clinical practice advances, clinicians are increasingly recognizing that dynamic monitoring of the MET gene can facilitate real-time adjustments to treatment regimens. This includes the timely initiation of MET inhibitor therapy upon detection of MET gene amplification or mutation, thereby enhancing overall patient survival and quality of life. One approach to achieve dynamic monitoring of MET status is through the use of liquid biopsies [71]. Traditionally, tissue biopsies (e.g., tumor sections) have been the primary source for detecting MET protein expression and gene changes for IHC, FISH, and NGS assays. Liquid biopsies encompass the extraction of circulating tumor DNA (ctDNA), exosomes, or circulating tumor cells (CTCs) from biological fluids, particularly blood, followed by the identification of MET alterations utilizing hybrid capture digital next-generation sequencing technology [72]. The collection of multiple samples enables the real-time monitoring of dynamic tumor changes, including therapeutic resistance and tumor recurrence, thereby offering a more comprehensive molecular profile compared to traditional tissue biopsies. A significant challenge in integrating liquid biopsies into clinical practice lies in the standardization of the testing process and the interpretation of results, as operational discrepancies between laboratories can result in inconsistent outcomes. Furthermore, the integration of liquid biopsy results into clinical treatment decisions necessitates additional research and validation. With technological advancements and the increasing adoption of liquid biopsy, more personalized and dynamic MET testing is anticipated in the future. This progression will substantially enhance patient management, provide more precise treatment options, and may play a key role in early detection and prevention of tumor recurrence.

### 3.2. Role in NSCLC Development

Dysregulation of HGF/c-MET signaling in NSCLC is associated with tumor proliferation, angiogenesis, invasion, metastasis, and acquired drug resistance. Furthermore, MET signaling is capable of interacting with other signaling pathways, such as the EGFR pathway, enhancing the complexity of NSCLC pathogenesis and treatment resistance.

HGF is secreted by stromal cells in the tumor microenvironment, and binds to and activates c-MET receptors on the surface of NSCLC cells, initiating a series of downstream signaling pathways such as PI3K/AKT/mTOR, MAPK, and Wnt/β-catenin. These pathways play critical roles in tumorigenesis and progression. For instance, the PI3K/AKT/mTOR pathway regulates NSCLC proliferation, differentiation, and apoptosis, and is also involved in resistance to EGFR-TKIs [73]. MiR-34a/miR-206 target c-MET, and knockdown of those miRNAs can regulate resistance to targeted therapies via the PI3K/AKT pathway [74,75]. Ying et al. showed that CAF-secreted HGF induces the activation of GRP78 and the PI3K/AKT pathways in NSCLC, resulting in resistance to paclitaxel [76]. The MAPK pathway is a central regulator of cell proliferation, differentiation and survival, promoting cell overgrowth by activating proliferation genes. Meanwhile, MAPK signaling inhibits AMPK signaling, one of the key metabolic nodes in cells, allowing them to overcome metabolic stress [77]. The HGF/c-MET axis is associated with Wnt/β-catenin signaling, with studies indicating that Wnt/β-catenin signals Snail1 and Zeb1 regulate bone metastasis in lung cancer [78].

The HGF/c-MET pathway also regulates epithelial–mesenchymal transition (EMT) in NSCLC, an important process that enhances tumor cell migration and invasion, and serves as an adaptive drug resistance mechanism [79]. HGF/c-MET signaling promotes altered cell morphology and modulation of intercellular adhesion molecule expression, facilitating NSCLC adaptation to the surrounding environment. Additionally, activation of HGF/c-MET signaling has been shown to enhance the cancer stem cell properties of NSCLC [80].

Furthermore, the interplay between c-MET and other receptors in promoting non-small cell lung cancer (NSCLC) progression has garnered increasing attention. Heterodimers of MET with EGFR, HER2, HER3, or RET exhibit distinct roles in tumor development. In MET-amplified NSCLC, heterodimers of MET with EGFR, HER2, and HER3 activate the AKT and ERK signaling pathways, thereby enhancing lung cancer cell proliferation and survival. Conversely, heterodimers of MET with HER2 and RET activate STAT3 signaling, which promotes cell migration [30]. The ligand-dependent heterodimerization of the HER family and c-Met has been extensively demonstrated to activate signaling pathways such as Src, PI3K-AKT-mTOR, and Ras-MAPK, contributing to the invasiveness and metastatic potential of tumor cells [34,81]. Additionally, the c-MET/VEGFR-2 heterodimeric complex inhibits VEGF by activating c-MET signaling, thereby promoting a more aggressive and metastatic phenotype [82]. An increasing volume of research indicates that the formation of heterodimers between distinct receptors is frequently linked to resistance against targeted therapies. This resistance arises because heterodimers can activate intracellular signaling pathways in a distinctive manner [83,84]. For instance, in NSCLC cells resistant to EGFR-TKIs, MET is significantly amplified and associates with HER3. MET functions as a chaperone for HER3 dimerization, facilitating HER3 phosphorylation and consequently contributing to drug resistance. The kinase activity of MET is essential for the formation of MET-HER3 heterodimers [85]. A study shows that MET overexpression/overactivation leads to NSCLC resistance to dual EGFR/Her2 inhibitors due to physical interaction of Met with EGFR and Her2 in NSCLC [86].

Crosstalk between c-MET and EGFR pathways in NSCLC has been well characterized. Studies have shown that c-MET gene amplification and c-MET protein overexpression are related to EGFR-TKI resistance in NSCLC. Approximately 5–20% of NSCLC patients exhibit c-MET amplification after EGFR-TKI treatment, which is closely related to acquired drug resistance [87,88]. Mechanistically, when c-MET is amplified or overexpressed, it can bypass EGFR pathway inhibition through the ERBB3 receptor-mediated PI3K/AKT and Grb2/MAPK pathways, thereby maintaining tumor cell growth and survival [85].

### 3.3. Drug Development of MET Inhibitors

In NSCLC, targeting the HGF/c-MET signaling pathway is of great clinical importance. Currently, several MET inhibitors, including small molecule tyrosine kinase inhibitors (TKIs) and monoclonal antibodies, have demonstrated promising anti-tumor activities in clinical trials.

MET TKIs are a class of targeted therapeutic agents that address mutations or amplifications of the MET gene. On the basis of their structure and binding mechanism to MET, they can generally be classified into three categories. This type of classification is useful for a better understanding of their mechanisms of action and their potential mechanisms of resistance. Clinical trials have primarily examined type I and type II ATP-competitive inhibitors. Type I inhibitors fill the ATP-binding pocket by interacting with residue Y1230 and are further divided into Type Ia and Type Ib. Type Ia inhibitors belong to non-selective MET inhibitors, such as crizotinib, that bind MET non-selectively via the solvent-terminal residue G1163 [65]. Type Ib inhibitors, including capmatinib, tepotinib, and savolitinib, bind selectively to MET independently of residue Y1230. By precisely inhibiting MET, Type Ib MET TKIs exhibit higher efficacy and better patient tolerability [89]. Type II MET TKIs, such as cabozantinib, merestinib, tivantinib, and glesatinib, are ATP-competitive inhibitors, occupying the ATP-binding pocket and binding to MET in its inactive state (DGF-out). Type III MET TKIs, in contrast, are non-ATP-competitive inhibitors that bind outside the ATP structural domain [90]. The efficacy of MET TKIs in treating NSCLC has been studied in several clinical trials (Table 1).

In addition to MET TKI, antibodies targeting HGF/c-MET and antibody–drug conjugates (ADCs) have been investigated for the treatment of patients with MET-altered NSCLC. There are anti-met antibodies, such as onartuzumab and emibetuzumab, and anti-HGF antibodies, such as rilotumab and ficlatuzumab. While onartuzumab and emibetuzumab fail to show significant efficacy in clinical trials [103,104], novel bispecific antibodies like amivantamab, which have higher MET affinity, have demonstrated efficacy in patients with EGFR exon 20 insertion mutated NSCLC by overcoming TKI resistance. Amivantamab has received accelerated approval from the FDA, and multiple clinical trials have confirmed its efficacy and safety [105]. Additional bispecific antibodies, such as REGN5093 and Sym015, have demonstrated efficacy and a favorable safety profile in preliminary trials [106,107]. Antibody–drug conjugates (ADCs) like telisotuzumab vedotin and REGN5093-M114 have the potential to enhance anti-tumor activity and minimize toxicity through targeted antigen delivery of potent chemotherapeutic agents. Initial findings from ongoing clinical investigations of these ADCs are promising [108,109]. In conclusion, while these innovative antibodies and ADCs show promise as therapeutic options, further clinical research is necessary to confirm their long-term effectiveness and safety. Table 2 lists published or ongoing clinical trials of MET antibodies and MET ADCs in NSCLC.

## 4. HGF/c-MET Signaling in Microenvironment

The tumor microenvironment encompasses tumor cells and their surrounding interacting cells and tissues, which undergo continuous alterations and remodeling throughout the progression of cancer. It is composed of various components, such as immune cells, endothelial cells, adipocytes, blood vessels, and extracellular matrix, all spatially situated in close proximity to the cancer cells [42,118,119]. The dynamic interactions between tumor cells and the TME play a crucial role in tumor survival and progression, while also hindering immune system surveillance of tumor progression [120,121,122]. HGF is abundant in the TME and is mainly produced and secreted by tumor-associated fibroblasts (CAFs) in addition to being secreted by cancer cells themselves [123,124]. Simultaneously, the receptor c-MET is present in epithelial cells, and activation of the HGF/c-MET axis serves various functions within the TME, such as promoting metastatic characteristics, facilitating tumor angiogenesis, impacting cellular metabolism, mediating interactions between tumor cells and stromal cells, and suppressing the immune system’s ability to respond to tumor aggression through diverse mechanisms, ultimately aiding in tumor immune evasion (Figure 2).

### 4.1. Promotes Angiogenesis

HGF activates downstream signaling pathways such as PI3K/AKT and MAPK/ERK to stimulate endothelial cell proliferation and migration, which is an important step in angiogenesis [125,126,127]. Additionally, these pathways are capable of inducing the expression of matrix metalloproteinases (MMPs) to degrade the extracellular matrix, further supporting neovascularization. Collectively, HGF/c-MET signaling plays a pivotal role in the formation and maturation of new blood vessels, ultimately facilitating the essential blood supply required for tumor progression and metastasis [128]. VEGF plays a crucial role in angiogenesis, with its upregulation in tumor cells under hypoxic conditions leading to synergistic effects with HGF in promoting endothelial cell proliferation, migration, and vascular structure formation, thereby facilitating tumor angiogenesis [129]. The resistance to VEGFR inhibitors, aimed at blocking VEGF signaling to inhibit angiogenesis, poses a significant obstacle in cancer therapy [130]. Research has shown that stromal-derived HGF acts as a mediator of resistance to VEGFR TKIs, resulting in increased tortuosity and disorganization of the tumor vasculature [131]. This study established a mouse lung adenocarcinoma xenograft model, which was initially sensitive to VEGFR TKIs, but later developed resistance, as evidenced by tortuous tumor vasculature, and increased expression of stromal-derived HGF and activation of c-MET detected in cancer cells and tumor-associated mesenchyme. Inhibition of the VEGFR and c-MET signaling pathways delayed the emergence of resistance and prevented changes in vascular morphology [131].

### 4.2. Effects on Tumor Metabolism

It is well known that one of the hallmarks of tumor cells is metabolic reprogramming. Oncogene-driven tumorigenesis results in a metabolic shift in cancer cells towards aerobic glycolysis, known as the Warburg effect, as opposed to reliance on mitochondrial oxidative phosphorylation [132]. The MET signaling pathway has been shown to facilitate the development of a glycolytic phenotype in cells, resulting in increased lactate secretion [133]. Additionally, inhibition of Met using a MET inhibitor (PHA-665752) has been found to decrease the expression of glycolysis initiation-associated hexokinase 2 (HK2) and promote the reactivation of oxidative phosphorylation in cancer cells within a NSCLC model [134].

### 4.3. Interaction with Cancer-Associated Fibroblasts

The interaction between lung cancer cells and cancer-associated fibroblasts (CAFs) occurs via the HGF/c-Met signaling pathway, whereby lung cancer cells stimulate CAFs to release HGF, leading to the activation of c-MET and subsequent promotion of lung cancer cell proliferation and survival. This process establishes a positive feedback loop [135]. Mechanistically, tumor cells may induce CAFs to produce HGF through the production of lactate. Notably, activation of the HGF/c-Met signaling pathway can upregulate glycolysis, resulting in increased lactate secretion from cancer cells [136,137].

### 4.4. Modulating Immune System and Anti-Tumor Immunity

Immune infiltration is also a significant factor influencing changes in the tumor microenvironment. The HGF/Met signaling pathway not only plays a role in modulating immune infiltration within the TME, but also impacts tumor cells in evading the anti-tumor immune response [138,139,140,141]. Infiltrating immune cells are integral to the processes of tumor growth and modulation of anti-tumor immune responses. For instance, cytotoxic T-lymphocytes and natural killer (NK) cells are involved in inhibiting tumor growth, while tumor-associated macrophages (TAMs) and tumor-associated neutrophils may promote tumor progression. Regulatory T-cells and myeloid-derived suppressor cells (MDSCs) are known to suppress anti-tumor immunity [142]. In fact, the initial infiltration of immune cells is beneficial for tumor control, but the interaction between tumor cells and immune cells within the TME ultimately creates a conducive environment for tumor progression and spread [143]. Further exploration of the HGF/Met signaling pathway’s involvement in this interplay is crucial for elucidating the intricate mechanisms of tumor biology and laying the groundwork for novel therapeutic approaches to enhance the effectiveness of immunotherapy and the outcomes of cancer patients.

Mature T cells are divided into two subsets: CD4+ and CD8+. Cytotoxic CD8+ T cells (CTL) play a more central role in anti-tumor immunity than CD4+ T cells. When stimulated by processed tumor antigen-derived peptides, naïve CD8+ T cells differentiate into effector CD8 + T cells and are further activated into cytotoxic CD8+ T cells and memory CD8 + T cells to exert their targeting effects at the tumor site [144]. Activated CD8+ T cells directly kill tumor cells by releasing perforin and granzymes and express death ligands, mainly Fas ligand (FasL) and TNF-related apoptosis-inducing ligand (TRAIL), which induce apoptosis of tumor cells through death ligand/death receptor binding [145]. It is reported that c-MET can be recognized as a tumor-associated antigen (TAA) by CD8+ T cells, triggering the activation of the immune system against MET-overexpressing cancer cells [146]. This suggests that MET can act as TAA to exert anti-tumor effects. However, in most cases, HGF/c-MET signaling inhibits CTL-mediated tumor killing. CTLs themselves have been reported to express c-MET, and one study identified a population of c-Met+ CTLs with high solubilizing activity in a mouse lung metastasis model [147,148]. Whereas HGF treatment inhibits its high lysogenic activity by reducing GrB and IFN-γ production, targeting the HGF/c-Met pathway enhances its mediated antitumor immunity [149]. In addition, a study related to liver cirrhosis found that c-Met binds to HGF and also binds to Fas ligand to form a complex that induces apoptosis in CTLs, an effect that may promote cancer progression [150]. Similarly, in an in vitro model of cytotoxic T-cell-dependent immunity, HGF was found to reduce CD8+ T-cell production, inhibit the secretion of cytokines and cytolytic enzymes (e.g., interferon (IFN)-γ, tumor necrosis factor (TNF), perforin, and granzyme B) and further reduce the expression of Fas ligand for membrane-bound death receptor (FasL), which greatly impaired the CD8+ T cell killing effect of CD8+ T cells [151]. As we mentioned before, HGF/c-Met signaling promotes a glycolytic phenotype in cancer cells, causing them to secrete more lactic acid, and the increased lactic acid inhibits CTL proliferation and activity [152]. In conclusion, inhibition of c-MET can increase the number of tumor-infiltrating T cells, enhance their activity, and improve the efficacy of immunotherapy [153].

In addition to regulating T cells, HGF/c-MET signaling also regulates the function and infiltration of immunosuppressive cells. These cells include tumor-associated neutrophils, tumor-associated macrophages, tumor-associated dendritic cells, myeloid-derived suppressor cells, and regulatory T cells (Tregs). They inhibit the immune response against tumor cells and facilitate tumor immune escape. As crucial components of the tumor immunosuppressive microenvironment, these cells impede the efficacy of antitumor therapy [154].

A feature of many cancers is a large infiltration of neutrophils, which may have antitumor (N1) or pro-tumor (N2) effects. Activated neutrophils have been reported to release HGF [155]. Granulocyte-macrophage colony-stimulating factor (GM-CSF) has been suggested to be a key determinant in the induction of HGF production by neutrophils from malignant cells in vitro and in vivo [156]. Further studies have confirmed that neutrophil-derived HGF plays a pro-tumorigenic role. In bronchoalveolar carcinoma (BAC), neutrophil infiltration and HGF secretion are associated with poor prognosis [157]. In addition, Cadranel and colleagues found that bronchoalveolar lavage fluid from BCA patients contained a high concentration of biologically active HGF, which promoted the migration of BAC tumor cells [157]. Analysis of clinical data has shown that high levels of serum HGF are associated with an increase in the number of neutrophils and with adverse effects of immune checkpoint blockade therapy. This is an indication that MET may be a potential target for immunotherapy. Glodde et al. showed that even in tumors that do not rely on c-MET, interference with the c-MET tyrosine kinase receptor attenuates neutrophil-dependent immunosuppression and enhances the efficacy of immunotherapy [153]. Additionally, recent research has shown that inhibiting neutrophil or MET signaling, in combination with ICB, can hinder bladder cancer progression and bolster the antitumor response of mouse CD8+ T cells [158]. In contrast, another study showed that activation of the HGF/c-MET signaling pathway in neutrophils has antitumor effects. In fact, MET is also expressed in neutrophils. This expression is induced by tumor-derived tumor necrosis factor (TNF)-α or inflammatory stimuli. Activation of c-MET on the surface of neutrophils by HGF leads to chemotactic attraction to tumors and the release of nitric oxide to kill tumor cells, whereas conditional deletion of the MET gene in neutrophils promotes tumor growth and spread to other organs [159]. Despite the dual roles of HGF/c-Met signaling in neutrophils—both anti-tumor and pro-tumor—the pathway predominantly exerts an immunosuppressive effect on the anti-tumor immune response. A recent study found that overexpressing c-MET in cancer cells increased the secretion of CXCL1/2, G-CSF, and GM-CSF—immune factors that are crucial for neutrophil recruitment, granulogenesis, and homeostatic function [108]. Additionally, the conditioned medium from c-MET-overexpressing cells significantly induced neutrophils to secrete lipocalin 2 (LCN2), which in turn promoted the self-renewal of cancer stem cells, a process reported to induce immunosuppression [160,161].

Studies have shown that HGF regulates monocyte function in a paracrine/autocrine manner. HGF induces directional migration of monocytes and secretion of cytokines such as GM-CSF, G-CSF, and IL-6 [162]. Additionally, HGF produced by mesenchymal stem cells (MSCs) induces monocytes to acquire an immunomodulatory phenotype, suppressing T-cell function and promoting high levels of IL-10 production via the ERK1/2 pathway [163]. In the TME, a variety of chemokines, cytokines, and soluble metabolites control the recruitment of immune cells to and within tumors. HGF recruits and activates tumor-associated macrophages (TAMs) [139]. TAMs in the TME are classified into M1 (anti-tumor) and M2 (pro-tumor) types. M1 macrophages secrete pro-inflammatory cytokines with anti-tumor effects, while M2 macrophages induce anti-inflammatory signals that promote tumor progression [164]. HGF promotes the transition of macrophages to the M2 type [165]. A recent study found that M2 macrophages also secrete HGF to maintain tumor growth and metastasis. They use chemotaxis to attract more macrophages from surrounding tissues, regulating M2 macrophage distribution and increasing hepatocellular carcinoma’s resistance to sorafenib [166]. In glioblastoma (GBM), MET overexpression was associated with increased macrophage numbers and poor prognosis [167].

In the immune system, HGF is a potent immunomodulatory factor that promotes dendritic cell (DC) adhesion and migration. HGF also acts as an immune tolerance factor by inducing DC differentiation into tolerogenic dendritic cells (DCregs), which further induce immune tolerance and anergy in T cells [168]. Further studies showed that dendritic cells (DCs) expressing c-MET and HGF treatment effectively inhibited the ability of DCs to present antigens (Ag) in vitro and in vivo, and downregulated Ag-induced Th1-type and Th2-type immune responses [169]. Mechanistically, the PI3K/AKT pathway plays a key role in mediating the inhibition of dendritic cell (DC) activation by HGF, and activation of Bruton’s tyrosine kinase (Btk) is required for this process [170,171]. Additionally, HGF may impair DC function by inducing the release of IL-10 from mature DCs and upregulating the expression of indoleamine 2,3-dioxygenase 1 (IDO1) in DCs. This process inhibits T-cell clonal expansion, suggesting that HGF participates in the remodeling of the immunosuppressive microenvironment by affecting DC function [172]. IL-10 impairs DC function by inhibiting their differentiation from stem cells or precursor cells and inducing apoptosis. This action of IL-10 has been shown to protect tumor cells from cytotoxic CD8+ T cells (CTLs) in the TME [173,174].

Myeloid-derived suppressor cells (MDSCs) are a heterogeneous population of bone marrow-derived cells consisting of myeloid progenitor cells and immature macrophages, immature granulocytes, and immature dendritic cells [175]. They play a crucial role in immunosuppression and represent a significant barrier to successful immunotherapy in NSCLC [119]. Mesenchymal stem cells (MSCs) have been reported to expand MDSCs by secreting HGF, which binds to the c-MET receptor, mediating downstream phosphorylation of STAT3 [124]. Expanded MDSCs express inducible nitric oxide synthase (iNOS) and arginase 1 (ARG1), leading to an increase in the number of Treg cells [176]. Activated MDSCs inhibit the expansion and function of cytotoxic T lymphocytes (CTLs) and further expand the immunosuppressive Treg population. This process involves mechanisms such as increased expression of Arg1 and iNOS, decreased reactive oxygen species (ROS) expression, and induction of Treg cells [177,178].

Regulatory T (Treg) cells expressing FOXP3 play a critical role in suppressing aberrant immune responses to self-antigens and anti-tumor immune responses [179]. It has been reported that mesenchymal stem cells (MSCs) induce the transformation of fully differentiated Th17 cells into functional Treg cells by secreting HGF [180]. In an autoimmune encephalitis (EAE) model, HGF induced an increase in CD25+FOXP3+ regulatory T (Treg) cells, which exerted anti-inflammatory effects and protected against EAE and multiple sclerosis [181]. In the tumor microenvironment, indoleamine 2,3-dioxygenase 1 (IDO1), a key regulator of T cell function, degrades tryptophan (Trp) to kynurenine (KYN), thereby inhibiting the function of effector T cells and promoting the differentiation and infiltration of immunosuppressive Treg cells [182]. A study found that in multiple myeloma (MM), high levels of serum HGF upregulated IDO1 expression through activation of AKT phosphorylation. This upregulation promoted Treg expansion, inhibited Th1 differentiation, and helped tumors evade T cell-mediated immune attack [183].

Within the immune microenvironment, there have been reports of an interaction between programmed death ligand 1 (PD-L1) and the HGF/c-MET pathway. PD-L1, also recognized as B7-H1 or CD274, serves as an immune checkpoint molecule that binds to programmed death-1 (PD-1) receptors on activated T-cells. This interaction results in T-cell exhaustion and immunosuppression, ultimately facilitating cancer cell immune tolerance and survival [184]. Immunosuppressive agents targeting PD-1/PD-L1 have shown clinical and preclinical benefits in NSCLC [185]. Numerous studies have indicated that the HGF/c-MET pathway plays a role in immunosuppression by sustaining PD-L1 expression through various pathways. Studies have shown that overexpression or activation of MET in NSCLC is linked to an increase in PD-L1 expression and the upregulation of immunosuppressive genes such as PDCD1LG2 and SOCS1 [186,187]. Mechanistically, c-MET amplification triggers PD-L1 expression through the PI3K/Akt and MAPK signaling pathways, leading to tumor immune evasion [188]. Furthermore, analysis of 18,047 NSCLC tumors demonstrated a higher prevalence of PD-L1 positivity in the MET mutant subgroup compared to the MET wild-type subgroup [189]. In addition to NSCLC, research has indicated that the activation of the HGF/c-MET pathway is associated with increased PD-L1 expression in hepatocellular carcinoma following cisplatin treatment, downstream of the PI3K/Akt and MET/ERK pathways [190]. Likewise, c-MET has been shown to influence PD-L1 transcription via the MAPK/NF-κBp65 pathway, contributing to the progression of hepatocellular carcinoma [191]. A study revealed a significant upregulation and co-localization of c-Met and PD-L1 in human renal cancer tissues. Activation of c-Met was found to induce PD-L1 expression through the Ras-PI3K pathway, providing protection to cancer cells against immune cell-mediated killing [192]. Furthermore, it has been proposed that elevated MET expression is linked to a negative prognosis and increased PD-L1 expression in pancreatic cancer. Additionally, MET has been shown to facilitate PD-L1 transcription through STAT1 and maintain PD-L1 stability in a CMTM6-dependent manner [193]. In primary glioblastoma (GBM), the MET-STAT4-PD-L1 axis has been identified as promoting tumor immune evasion and correlating with a poor prognosis [167].

A study discovered that MET in MET-amplified EGFR-TKI-resistant cells inhibits STING-induced tumor immunogenicity by increasing CD73 expression, serving as a novel immune checkpoint and potential therapeutic target to enhance extracellular adenosine (eADO) production. This process ultimately hinders the activity of T cells and natural killer cells, resulting in an immunosuppressive environment within the tumor [194,195,196].

## 5. Targeting HGF/c-Met Signaling Combined with Immunotherapy in NSCLC

### 5.1. The Significance of Targeting MET Combined with Immunotherapy

Recent advancements in immunotherapy for NSCLC have been notable, particularly with the emergence of immune checkpoint inhibitors (ICIs) such as PD-1/PD-L1 inhibitors. The combination of MET targeted therapy with immunotherapy presents promising opportunities in the realm of cancer treatment. Significantly, dysregulation of c-MET in non-small cell lung cancer (NSCLC) not only facilitates tumor proliferation, angiogenesis, and metastasis, but also alters the immune microenvironment. As previously elucidated, c-MET hinders cytotoxic T lymphocyte (CTL) activity, attracts immunosuppressive cells, stimulates the secretion of immunosuppressive factors, and upregulates PD-L1 expression, thereby impacting the effectiveness of immunotherapy. Hence, the concurrent utilization of MET-targeted therapy and immunotherapy may yield a synergistic outcome, offering a more efficacious and holistic treatment approach for individuals with cancer. A research. demonstrating that inhibition of MET in neutrophils enhances tumor progression, introduces the concept of integrating MET-targeted therapy with immunomodulatory agents for the first time [159]. Numerous preclinical and clinical studies have provided substantial evidence endorsing the combination of MET-targeted therapies with immunotherapy.

It is widely recognized that if tumor cells express high levels of PD-L1, blocking PD-L1 is essential for activating anti-tumor T cells and promoting anti-tumor immunity [197]. A research study demonstrated that the use of the c-MET inhibitor tivantinib resulted in an increase in PD-L1 expression in NSCLC cell lines, leading to evasion of T-cell-mediated killing by cancer cells. This suggests a potential interaction between c-MET inhibition and immune evasion, providing a theoretical basis for combining a c-MET inhibitor with immune checkpoint blockade in the treatment of NSCLC [198]. Additionally, in liver cancer, c-MET inhibitor was found to stabilize PD-L1, and the combined therapy of c-MET inhibitor capmatinib with anti-PD-1 therapy showed a significantly improved therapeutic outcome [199]. In NSCLC, MET and PD-L1 expression are positively correlated. An analysis of clinical data indicated a notable increase in PD-L1 expression in MET-amplified NSCLC, suggesting that MET-amplified tumors may benefit from anti-PD-L1 therapy [200]. Beyond MET amplification, NSCLC patients with MET non-exon 14 mutations showed favorable outcomes with immune checkpoint inhibitors [151,152]. Conversely, NSCLC patients with MET exon 14 alterations (METex14) demonstrate a lower overall response rate to immune checkpoint blockade and a shorter median progression-free survival, despite the majority of patients expressing PD-L1 [201].

Previous studies have demonstrated that immune checkpoint inhibitor (ICI) therapies have significantly impacted the treatment of NSCLC [202]. However, MET has been identified as a key biomarker and driver mediating resistance to ICIs in NSCLC. Zhang discovered that patients with MET amplification exhibited resistance to ICIs, resulting in decreased progression-free survival. Subsequent investigations revealed that tumors with MET amplification displayed reduced levels of STING and less infiltration of CTLs and NK cells, and decreased tumor immunogenicity of the MET-amplified tumor microenvironment was confirmed by single-cell RNA sequencing. Mechanistically, oncogenic MET signaling leads to reduced STING expression by inducing UPF1 phosphorylation. Encouragingly, inhibition of MET can overcome the problem of reduced ICB efficiency due to MET amplification [203]. In other tumors such as renal cell carcinoma, MET inhibition has also been identified as a key factor in overcoming resistance to immune checkpoint inhibitors and maximizing the efficacy of immunotherapy [204].

### 5.2. Current Therapeutic Strategies

It has been established that MET plays a crucial role in mediating resistance to monotherapy with immune checkpoint inhibitors (ICIs), and the combination of MET inhibitors with anti-PD-L1 therapies may exhibit heightened efficacy in the treatment of NSCLC. Regrettably, based on the clinical trils reported in NCT04139317 and NCT04323436 www.clinicaltrials.gov (accessed on 16 August 2024), the combination of MET TKI and ICI for NSCLC has demonstrated an elevated incidence of adverse events and inadequate patient tolerability. It is imperative to implement precise dosage adjustments and management protocols to mitigate the issue of tolerability. This integrated approach has produced significant benefits in other cancer types like pancreatic cancer and renal cell carcinoma [193,204]. The specific results of the clinical trials are shown in Table 3. 

The utilization of bispecific antibodies (BsAbs) targeting METs and immune molecules represents a promising strategy in cancer therapy. Through their dual binding capacity to specific antigens on tumor cells and receptors on immune cells, BsAbs facilitate the formation of a functional bridge between these two cell types, ultimately enhancing the immune system’s ability to recognize and eliminate cancerous cells. BsAb targeting c-MET and PD-1 significantly inhibited NSCLC growth [205]. Li and colleagues developed and produced a novel bispecific antibody, BsAb-5, that effectively targets c-MET and CTLA-4 in lung cancer stem cell (LCSC)-like cells with a high degree of specificity and affinity. This antibody demonstrated significant antitumor effects both in vitro and in vivo by inhibiting HGF-mediated tumor growth and enhancing anti-tumor T cell activity [206]. In addition, Lei et al. successfully synthesized the c-Met/CD3 bispecific antibody BS001, which not only effectively mediated tumor cell killing by T cells by bridging CD3-positive T lymphocytes and tumor cells, but also inhibited c-Met signaling and showed significant efficacy in combination with PD-L1 antibody [207]. In the context of gastric cancer, novel bispecific antibodies such as PD-1/c-Met DVD-Ig and IgG-scFv have been developed to target PD-1 and c-Met, leading to restoration of T-cell function, suppression of tumor growth, and promising implications for the treatment of gastric cancer and other malignancies [208]. However, these novel agents are still in the preclinical modelling stage.

Recent advancements have been made in MET chimeric antigen receptor (CAR) T-cell immunotherapy, which involves modifying a patient’s own T-cells to better target and eliminate tumor cells [209]. This therapy, specifically targeting MET, has shown promising results in enhancing the anti-tumor activity of CAR-T cells and overcoming immunosuppression in solid tumor microenvironments. It is anticipated to be a novel therapeutic approach for the treatment of MET-overexpressing NSCLC [209]. Min et al. have demonstrated that CAR-T cells targeting c-Met exhibit heightened efficacy against NSCLC both in vitro and in vivo [210]. Thayaparan et al. found that MET-targeted CAR-T cells induce tumor regression through an immune response, leading to the release of IFN-gamma and IL-2 [211]. Yuan X et al. have engineered PD-1 inhibition-resistant bifunctional CAR-T cells that target c-Met and block the PD-1/PD-L1 interaction, effectively suppressing the growth of PD-L1-positive tumors [212]. In a similar manner, Wei Jiang and colleagues developed bispecific c-Met/PD-L1 CAR-T cells that demonstrated superior anti-hepatocellular carcinoma activity compared to monospecific c-Met or PD-L1 CAR-T cells [213]. This development of MET CAR-T represents an innovative approach for treating various solid tumors, exhibiting significant anti-tumor activity in papillary renal cell carcinoma (PRCC), recurrent nasopharyngeal carcinoma (rNPC), Ewing sarcoma (EWS), and gastric cancer (GC) [214,215,216,217]. A study evaluated the safety of MET-CAR immunotherapy at the preclinical level. In the study by Cristina et al., the cytotoxicity of MET-CAR T was evaluated in a variety of MET overexpression models, confirming that its killing effect was positively correlated with the level of MET expression and that it was essentially nontoxic to normal cells [218]. Limited data are available on the use of MET CAR T in evaluating CAR T therapy in solid tumor patients. Preliminary clinical data on the injection of MET CAR T cells into breast cancer tumors demonstrated favorable tolerability and induced necrosis, hemorrhage, and inflammatory cell infiltration within the tumor [219]. A phase I trial (NCT03060356) assessing the safety of intravenous RNA electroporation of c-MET-directed CAR T for metastatic breast cancer or melanoma revealed that six patients experienced grade 1 or 2 toxicity, with no instances of grade 3 or higher toxicities, neurotoxicity, or treatment discontinuation, indicating an overall safe treatment profile [112]. These findings support the promising potential of CAR T immunotherapy and suggest that further investigation through preclinical and clinical studies is warranted to evaluate its effectiveness and safety in various solid tumor types over the long term.

Additionally, the development of MET-targeted CAR NK cell immunotherapy has shown encouraging results. CAR NK cell therapy may have advantages over CAR T cell therapy in terms of reduced off-target effects and reduced immune-related toxicity. Yan Peng et al. successfully constructed c-Met-CAR-NK cells and designed DAP10, a favorable stimulatory factor for NK cell activation, into the CAR structure, which exhibited significant anti-tumor effects in a preclinical model of c-MET-positive lung adenocarcinoma (LUAD) [220]. Furthermore, Chutipa et al. successfully demonstrated the efficacy of anti-MET CAR-NK-92 cells in targeting and eliminating cholangiocarcinoma cancer cells, while Liu et al. utilized c-MET-targeted CAR NK cell immunotherapy for hepatocellular carcinoma, showing specific cytotoxicity against c-MET-positive HepG2 cells in vitro [221,222]. Subsequent studies may focus on refining the design of MET CAR NK cells, enhancing their targeting and cytotoxic abilities, and advancing clinical data to evaluate their safety and effectiveness.

## 6. Conclusions

Since the identification of MET as a driver gene in non-small cell lung cancer (NSCLC), research on the HGF/c-MET signaling pathway has significantly advanced. The pivotal role of the HGF/c-MET signaling pathway in promoting NSCLC progression, invasion, and metastasis is well recognized. Moreover, MET has emerged as a prominent mediator of immunotherapy resistance, particularly with the increasing use of immunotherapy in NSCLC treatment. However, the precise role of the HGF/c-MET signaling pathway within the tumor immune microenvironment remains incompletely understood. Recent studies have shown that the involvement of the HGF/c-MET signaling pathway within the tumor microenvironment (TME) is multifaceted and paradoxical. Specifically, HGF has the ability to attract neutrophils for anti-tumor activities [159], while MET serves as a tumor-associated antigen (TAA) in facilitating immune-mediated tumor destruction [146]. Conversely, the HGF/c-MET axis also recruits immunosuppressive cells, such as myeloid-derived suppressor cells (MDSCs) and regulatory T cells (Tregs), induces the production of immunosuppressive molecules like interleukin-10 (IL-10), and hinders the cytotoxic function of cytotoxic T lymphocytes (CTLs) [172,176]. Elevated MET expression is frequently correlated with unfavorable outcomes in immunotherapy, indicating its predominant function as an immunosuppressive factor in tumor immunity. Ongoing investigations are focused on elucidating the mechanisms underlying this association, including MET-induced enhancement of PDL1 expression and MET amplification’s inhibition of the STING pathway [198,203]. Furthermore, the interaction between c-MET and other receptors may contribute to the establishment of an immunosuppressive microenvironment, thereby facilitating tumor cells’ evasion of immune surveillance. This phenomenon warrants further investigation. Additionally, a comprehensive exploration of MET signaling’s role within the immune system and its effects on anti-tumor immune responses is imperative.

Given the immunosuppressive function of the HGF/c-MET signaling pathway within the tumor microenvironment and the promising outcomes observed in preclinical studies targeting MET in conjunction with immunotherapy, there is a growing interest in the development of combination immunotherapies targeting MET and subsequent clinical trials. Nevertheless, preliminary clinical findings from trials combining MET-TKI with anti-PDL1 therapy suggest inadequate patient tolerability, as a significant portion of participants exhibited disease or clinical progression, and nearly one-third experienced adverse effects. At present, there is a lack of additional data confirming the safety and effectiveness of combining MET-TKI and ICI therapies. Furthermore, emerging treatments like MET CAR-T and MET CAR-NK are gaining traction as more precise treatment options, but they may face obstacles in clinical implementation due to difficulties in selecting and targeting specific receptors. This is due to the widespread expression of MET in both malignant and non-malignant cells, which could result in off-target toxicity when targeting MET. Additionally, CAR-T and CAR-NK therapies have the potential to induce severe cytokine release syndrome (CRS) or neurotoxicity. The emergence of resistance is also inevitable, as tumor cells may evade CAR-T or CAR-NK cell attacks by downregulating or completely losing MET expression. Due to the significant variation in tumor characteristics and immune status among different patients, future research needs to develop personalized treatment plans and identify reliable biomarkers to accurately assess the function of the HGF/c-MET axis in each patient.

In summary, the pivotal role of the HGF/MET pathway in promoting invasion and metastasis, angiogenesis, and immune evasion in non-small cell lung cancer (NSCLC) highlights its significance in the disease process. Notably, numerous studies have demonstrated that the activation of the MET pathway plays a significant role in conferring resistance to cancer treatments, specifically emerging anti-PD-1/PD-L1 immunotherapies. This discovery underscores the potential of MET-targeted therapies and establishes a robust scientific rationale for the implementation of combination strategies or novel approaches, such as CAR-T cell therapy, in the treatment of NSCLC. These combination strategies hold the promise of overcoming the limitations of existing therapies, augmenting therapeutic effectiveness, and ultimately enhancing patient prognoses.

## Figures and Tables

**Figure 1 ijms-25-09101-f001:**
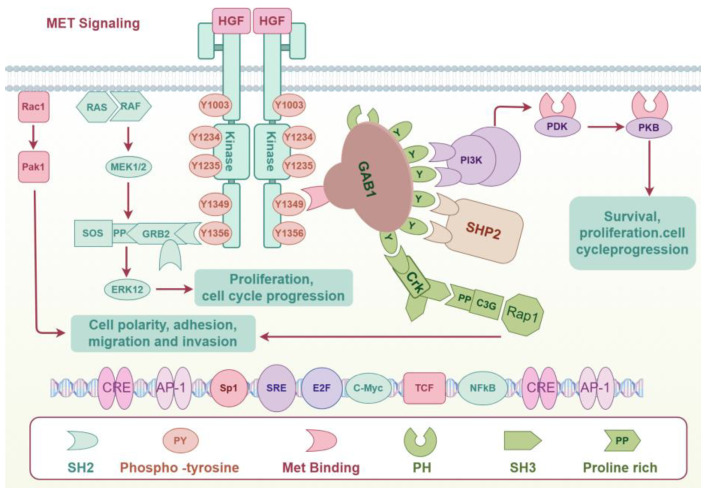
HGF/c-MET signaling.

**Figure 2 ijms-25-09101-f002:**
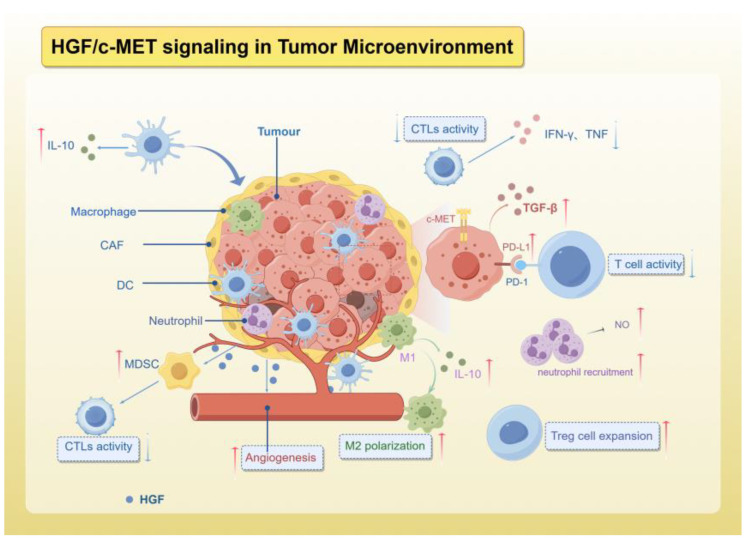
The role of HGF/c-MET signaling in TME.

**Table 1 ijms-25-09101-t001:** Preliminary activity of MET tyrosine kinase inhibitors in clinical trials for the treatment of MET-altered non-small cell lung cancer (NSCLC).

Drug	Trial	Phase	Treatment	Population	Methodological Platforms	N. of Patients	ORR	PFS	DOR	Ref.
Crizotinib	PROFILE 1001 NCT00585195	I	Crizotinib 250 mg b.i.d.	MET exon 14 skipping	NGS/RT-PCR	69	32%	7.3 months (95% CI, 5.4–9.1)	9.1 months (95% CI, 6.4–12.7)	[91]
	METROS NCT02499614	II	Crizotinib 250 mg b.i.d.	MET exon 14 skipping	Sanger direct sequencing	26	27%	4.4 months (95% CI, 3.0–5.8)	3.7 months (95% CI, 1.1–6.3)	[92]
	NCT00585195	I	Crizotinib 250 mg b.i.d.	MET amplification	NGS/RT-PCR	Low: 3 Medium: 14 High: 20	33% 14.3% 40%	1.8 months (95% CI, 0.8–14.0) 1.9 months (95% CI, 1.3–5.5) 6.7 months (95% CI, 3.4–7.4)	12.1 months (95% CI, 12.1–12.1) 3.7 months (95% CI, 3.7–3.7) 5.5 months (95% CI, 3.3–25.8)	[93]
	AcSé (NCT02034981)	II	Crizotinib 250 mg b.i.d.	c-MET ≥ 6 copies all c-MET-mutations	IHC/FISH/NGS	25 28	16% 10.7%	3.2 months (95% CI, 1.9–3.7) 2.4 months (95% CI, 1.6–5.9)	Not reported	[94]
Capmatinib	GEOMETRY mono-1 NCT02414139	II	Capmatinib 400 mg b.i.d.	MET exon 14 skipping MET exon 14 skipping MET amplification MET amplification	FISH/RT-PCR	28 69 69 15	68% 41% 29% 40%	12.4 months (95% CI, 8.2–NR) 5.4 months (95% CI, 4.2–7.0) 4.1 months (95% CI, 2.9–4.8) 4.2 months (95% CI, 1.4–6.90)	12.6 months (95% CI, 5.6–NR) 9.7 months (95% CI, 5.6–13.0) 8.3 months (95% CI, 4.2–15.4) 7.5 months (95% CI, 2.6–14.3)	[7]
	NCT01610336	II	Gefitinib 250 mg daily Capmatinib 400 mg b.i.d.	MET amplification copy number < 4	IHC/FISH	41	12%	Copy number ≥ 6 5.49 months (95% CI: 4.21–7.29) IHC: +3 5.45 months (95% CI: 3.71–7.10)	Not reported	[95]
				Copy number ≥ 4 < 6 Copy number ≥ 6 IHC: 0 IHC: +1 IHC: +2 IHC: +3		18 36 4 2 16 78	22% 47% 25% 0% 19% 32%			
Cabozantinib	NCT01866410	II	Cabozantinib 40 mg daily + erlotinib 150 mg daily	Advanced NSCLC with EGFR mutation and progressive disease on EGFR TKI (no MET mutations)	FISH	37	10.8%	3.6 months (95% CI, 2.0–5.6)	Not reported	[96]
	NCT01708954	II	Arm A: erlotinib 150 mg daily Arm B: cabozantinib 60 mg daily Arm C: erlotinib 150 mg + cabozantinib 40 mg	Previously treated advanced NSCLC(MET mutations not evaluated)	Not evaluated	38 38 35	3% 11% 3%	1.8 months (95% CI 1.7–2.2) 4.3 months (95% CI 3.6–7.4) 4.7 months (95% CI 2.4–7.4)	Not reported	[97]
Savolitinib	TATTON NCT02143466	Ib	Savolitinib 300 mg or 600 mg + Osimertinib 80 mg/daily	MET amplification(Post-1st/2nd-generation EGFR TKI T790M-)	FISH	51	65%	9.0 months (95% CI: 5.5–11.9)	9.0 months (95% CI: 6.1–22.7	[98]
			Savolitinib 300 mg + Osimertinib 80 mg/daily	MET amplification Post-1st/2nd-generation EGFR TKI T790M+ Post-3rd-generation EGFR TKI Post-1st/2nd-generation EGFR TKI T790M-		18 69 36	67% 21% 23%	11.0 months (95% CI: 4.0–NR) 5.4 months (95% CI: 4.1–8.0) 9.1 months (95% CI: 5.4–12.9)	12.4 months (95% CI: 2.8–NR) 7.9 months (95% CI: 4.0–10.5) 8.0 months (95% CI: 4.5–NR) 8.0 months (95% CI: 4.5–NR)	
	NCT02897479	II	Savolitinib 600 mg for BW ≥ 50 kgor 400 mg for BW < 50 kg	MET exon 14 skipping	NGS	61	49.2%	6.9 months	Not reported	[99]
	NCT02127710	II	Savolitinib 600 mg/daily	MET kinase domain mutant/amplified	FISH/NGS	41	18%	6.2 months (95% CI: 4.1–7.0)	Range: 2.4–16.4 months	[100]
				MET kinase domain mutant/amplified		65	0%	1.4 months (95% CI: 1.4–2.7)		
	NCT02897479	II	Savolitinib 600 mg/day	MET exon 14 skipping	NGS	34	38.7%	Not reported	34 weeks (range, 16–96)	[101]
Tepotinib	VISION NCT02864992	II	Tepotinib 500 mg/day	MET exon 14 skipping	NGS	87	BIRC: Liquid biopsy: 50% tissue biopsy: 45.1%	BIRC: liquid biopsy: 9.5 months (95% CI, 6.7–NR) tissue biopsy: 10.8 months (95% CI, 6.9–NR)	Not reported	[102]
							Inv: liquid biopsy: 55.3% tissue biopsy: 54.9% BIRC:	Inv: liquid biopsy: 9.5 months (95% CI, 5.3–21.1) tissue biopsy: 12.2 months (95% CI, 6.3–NR)		
Foretinib	NCT00726323	II	Foretinib	MET mutation MET amplification Chromosome 7 polysomy	IHC/FISH	74	13.5%	9.3 months (95% CI: 6.9–12.9)	18.5 months	

**Table 2 ijms-25-09101-t002:** Published or ongoing clinical trials of MET antibodies and MET ADCs in NSCLC.

Drug	Trial	Combination	Phase	Population	Methodological Platforms	Status *	Result	Ref.
Sym-015	NCT02648724	No	I/II	MET exon 14 skipping or MET amplification	Not evaluated	Completed	ORR 25% DCR 80% PFS 5.5 months (95% CI, 3.5–9.7)	[110]
Amivantamab	NCT04538664	Pemetrexed+Carboplatin	III	Advanced/Metastatic NSCLC Exon20 ins EGFR	Not evaluated	Active/Not recruiting	Superior efficacy of the combination versus chemotherapy alone (median PFS: 11.4 vs. 6.7 months; ORR: 73% vs. 47%)	[111]
	NCT04487080	Lazertinib	III	Advanced/Metastatic NSCLC Exon19 del or Exon 21 L858R EGFR	Not evaluated	Active/Not recruiting	Higher toxicity of the combination vs. monotherapy (≥Grade 3 AEs: 75% vs. 43%); Superior efficacy of the combination versus monotherapy (median PFS: 23.7 vs. 16.6 months)	[112]
	NCT04988295	Lazertinib+Pemetrexed+Carboplatin or Pemetrexed+Carboplatin	III	Advanced/Metastatic Non-squamous NSCLC Exon19 del or Exon 21 L858R EGFR; progressed on/after Osimertinib	Not evaluated	Active/Not recruiting	Higher toxicity of the Ami+Laze+chemo vs. Ami+chemo vs. chemo (≥Grade 3 AEs: 92% vs. 72% vs. 48%); Superior efficacy of the Ami+Laze+chemo or Ami+chemo versus chemo (median PFS: 8.3 vs. 6.3 vs. 4.2 months)	[113]
Teliso-V	NCT05513703	No	II	Advanced/Metastatic Non-Squamous NSCLC MET gene amplification	FISH	Active/Not recruiting	Not reported	
	NCT04928846	No	II	Previously Treated Non-Squamous NSCLC MET overexpression	IHC	Recruiting	Not reported	
	NCT02099058	None or Erlotinib or Nivolumab or Osimertinb	I	Advanced NSCLC	IHC	Active/Not recruiting	Safe and tolerated as monotherapy; antitumor activity in MET-positive patients Acceptable toxicity in combination with Erlotinib; encouraging antitumor activity in EGFR TKI pretreated/EGFR-mutated/MET-positive patients Tolerated in combination with Nivolumab; limited antitumor activity in MET-positive patients	[114,115,116,117]
	NCT06093503	Osimertinib	III	Advanced/metastatic non-squamous NSCLC MET overexpression	IHC	Not yet recruiting	N/A	
REGN5093-M114	NCT04982224	None or Cemiplimab	I/II	Advanced NSCLC MET overexpression	IHC	Recruiting	Not reported	

* Current status of clinical trials, information from https://clinicaltrials.gov/ (accessed on 16 August 2024).

**Table 3 ijms-25-09101-t003:** Clinical trials of combine MET inhibitors with anti-PD-L1 therapies, MET TKI.

Trial	MET TKI	ICI	Phase	Population	Status *	Result
NCT05782361	Tepotinib	Pembrolizumab	I	Advanced cancer/ NSCLC MET ex14 skipping positive	Recruiting	Not reported
NCT04139317	Capmatinib	Pembrolizumab	II	Advanced/metastatic NSCLC PD-L1 ≥ 50%	Terminated	Lack of tolerability of the combination
NCT04323436	Capmatinib	Spartalizumab	II	Advanced/metastatic NSCLC MET ex14 skipping positive	Terminated	Lack of tolerability of the combination
NCT03647488	Capmatinib	Spartalizumab	II	Advanced/metastatic NSCLC	Completed	The study was not opened in the randomized part 67% disease/clinical progression 27% Adverse events

* Current status of clinical trials, information from https://clinicaltrials.gov/ (accessed on 16 August 2024).

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
