# Peer review of "Unveiling the Role of HGF/c-Met Signaling in Non-Small Cell Lung Cancer Tumor Microenvironment"

_ijms, 2024, doi:10.3390/ijms25169101_

Round 1

Reviewer 1 Report

Comments and Suggestions for Authors

This review highlights the crucial role of the HGF/c-MET signaling pathway in NSCLC, emphasizing its involvement in tumor cell proliferation, migration, invasion, angiogenesis, and immune evasion. While the pathway's significance in NSCLC is well-established, its role in remodeling the tumor microenvironment remains underexplored. The review also discusses the current progress in MET-targeted therapies, suggesting the potential of novel MET inhibitors and combination immunotherapy. Despite these advancements, there are still significant gaps in understanding the pathway's broader impact on the tumor microenvironment. Addressing these gaps could enhance the effectiveness of MET-targeted treatments. Future research should focus on these unexplored areas to optimize therapeutic strategies. The review underscores the importance of c-MET as a promising target for inhibiting tumor progression but identifies critical areas needing further insights. 

Major Comments: 

1. The authors need to address the relationship between c-MET and other receptors, such as GPCRs and AXL, given recent advancements in understanding their roles in lung cancer progression.

2. The authors should differentiate between the roles of c-MET and pro-MET to clarify the mechanisms underlying c-MET activation.

3. The authors must discuss the heterodimerization of c-MET with other receptors and its impact on the tumor microenvironment.

Author Response

Comments 1: The authors need to address the relationship between c-MET and other receptors, such as GPCRs and AXL, given recent advancements in understanding their roles in lung cancer progression.

Response 1: Thank you for pointing this out. We agree with this comment. Therefore, We provide a comprehensive analysis of the relationship between the c-MET receptor and other receptors, including receptor tyrosine kinases (RTKs), AXL, and G-protein-coupled receptors (GPCRs). This includes examples of molecular mechanisms underlying the crosstalk between c-MET and these receptors.(page 3, paragraph 2 , and line 109-129.)

Comments 2: The authors should differentiate between the roles of c-MET and pro-MET to clarify the mechanisms underlying c-MET activation.

Response 2: Thank you for pointing this out. We agree with this comment. Therefore, we have incorporated an introduction to pro-MET and its processing into the mature c-MET receptor to elucidate the mechanism of c-MET amplification more effectively. (page 2, paragraph 5 , and line 86-87)

Comments 3: The authors must discuss the heterodimerization of c-MET with other receptors and its impact on the tumor microenvironment.

Response 3: Thank you for pointing this out. We agree with this comment. Therefore, We concur with the observation and subsequently conducted a comprehensive literature review on the effects of c-MET heterodimerization with other receptors in NSCLC.(page 6, paragraph 4 , and line 259-279)Our review indicates that existing studies predominantly emphasize its role in promoting tumor invasive metastasis and drug resistance, while there is a relative paucity of literature addressing its impact on the tumor microenvironment. In our conclusion, we explore the potential implications of c-MET heterodimerization with other receptors on the tumor microenvironment, drawing upon our understanding of the HGF/c-MET signaling pathway.(page 24, paragraph 1 , and line 703-706)

Reviewer 2 Report

Comments and Suggestions for Authors

The manuscript entitled Unveiling the Role of HGF/c-Met Signaling in Non-Small Cell 2 Lung Cancer Tumor Microenvironment represents a technically correct and timely relevant manuscript available for the publication on this journal after minor considerations

- In the introduction section, please, could the authors significantly add clinical data about MET implementation in clinical practice. As regards, I would kindly suggest overviewing the clinical trials supporting this aspect

- In the table 1 and 2, please, could the authors also add methodological platforms used to test MET? Accordingly, I would also recomend to discuss the post impacting technical strategies for MET analysis in clinical practice.

- Please, could the authors also discuss how biological samples may be relevant for MET testing in clinical practice? How liquid biopsy may be integrated in this vision? 

Comments on the Quality of English Language

Minor English editing

Author Response

Comments 1: In the introduction section, please, could the authors significantly add clinical data about MET implementation in clinical practice. As regards, I would kindly suggest overviewing the clinical trials supporting this aspect

Response 1: Thank you for pointing this out. We agree with this comment. Therefore, the introductory section, we have incorporated clinical data pertaining to the implementation of MET to support clinical practice.(page 1,2, paragraph 1 , and line 31-45)

Comments 2: In the table 1 and 2, please, could the authors also add methodological platforms used to test MET? Accordingly, I would also recomend to discuss the post impacting technical strategies for MET analysis in clinical practice.

Response 2: Thank you for pointing this out. We agree with this comment. Therefore, we have included methodological platforms for the detection of MET in Tables 1 and 2, and have elaborated on the methods for detecting changes in MET in clinical practice within the main text. (page 4, paragraph 3,4,5 and page 5, paragraph 1,2)Furthermore, we provide a discussion on advanced technical strategies that influence MET analysis in clinical practice.(page 5, paragraph 3, line 201-208)

Comments 3: Please, could the authors also discuss how biological samples may be relevant for MET testing in clinical practice? How liquid biopsy may be integrated in this vision? 

Response 3: Thank you for pointing this out. We agree with this comment. Therefore, We discuss MET assays for their application in clinical practice across various biological sample types, including tumor tissue and body fluids. Additionally, we evaluate the feasibility and challenge of integrating liquid biopsies into clinical practice.(page 5, paragraph 4, line 229-259)